# Locus coeruleus modulation of prefrontal dynamics during attentional switching in mice

**Marco Nigro[1], Lucas Silva Tortorelli[1], Machhindra Garad[1], Natalie Zlebnik[2,3], Hongdian Yang[1,3]***

[1]Department of Molecular, Cell and Systems Biology, University of California, Riverside, Riverside, United States; [2]Division of Biomedical Sciences, School of Medicine, University of California, Riverside, Riverside, United States; [3]Neuroscience Graduate Program, University of California, Riverside, Riverside, United States

## eLife Assessment

This study presents a **valuable** finding on how the locus coeruleus modulates the involvement of medial prefrontal cortex in set shifting using calcium imaging in mice. The evidence supporting the claims was viewed as **solid** in revealing the dynamics and potential mechanisms supporting extradimensional shifts. The work is of broad interest to those studying flexible cognition.

**\*For correspondence:**
hongdian@ucr.edu

**Abstract** Behavioral flexibility, the ability to adjust behavioral strategies in response to changing environmental contingencies and internal demands, is fundamental to cognitive functions. Despite a large body of pharmacology and lesion studies, the precise neurophysiological mechanisms that underlie behavioral flexibility are still under active investigations. This work is aimed to determine the role of a brainstem-to-prefrontal cortex circuit in flexible rule switching. We trained mice to perform a set-shifting task in which they learned to switch attention to distinguish complex sensory cues. Using chemogenetic inhibition, we selectively targeted genetically defined locus coeruleus (LC) neurons or their input to the medial prefrontal cortex (mPFC). We revealed that suppressing either the LC or its mPFC projections severely impaired switching behavior, establishing the critical role of the LC-mPFC circuit in supporting attentional switching. To uncover the neurophysiological substrates of the behavioral deficits, we paired endoscopic calcium imaging of the mPFC with chemogenetic inhibition of the LC in task-performing mice. We found that mPFC prominently responded to attentional switching and that LC inhibition not only enhanced the engagement of mPFC neurons but also broadened single-neuron tuning in the task. At the population level, LC inhibition disrupted mPFC dynamic changes and impaired the encoding capacity for switching. Our results highlight the profound impact of the ascending LC input on modulating prefrontal dynamics and provide new insights into the cellular and circuit-level mechanisms that support behavioral flexibility.

## Introduction

The ability to adjust behavioral strategies in response to changing external contexts and internal needs, termed behavioral/cognitive flexibility, requires adaptive processing of environmental cues and internal states to guide goal-oriented behavior, and is vital to the survival of organisms. Inappropriate behavioral adjustments, such as deficits in modifying responses to a rule change, are hallmarks

of impaired executive functions and are observed in a broad spectrum of psychiatric disorders (*Miller and Cohen, 2001*; *Uddin, 2021*).

Decades of research have strived to uncover the neural substrates of behavioral flexibility (e.g. see reviews *Miller and Cohen, 2001*; *Uddin, 2021*; *Mesulam, 1998*; *Miller, 1999*; *Ragozzino, 2007*; *Le Merre et al., 2021*). Set shifting, a type of rule switching that requires attending to or ignoring a stimulus feature in a context-dependent way, is commonly used to assess flexibility. The Wisconsin Card Sorting Test, the Intra-Extra Dimensional Set Shift Task, and their analogs have been widely used to test the ability of attentional switching in human and animal subjects (*Berg, 1948*; *Milner, 1963*; *Roberts et al., 1988*; *Dias et al., 1996a*; *Monchi et al., 2001*; *Barnett et al., 2010*; *Brown and Tait, 2016*; *Young et al., 2010*). Importantly, prior research using lesion and pharmacology approaches has provided compelling evidence that the medial PFC (mPFC) plays an important role in set shifting (e.g. *Ragozzino, 2007*; *Dias et al., 1996b*; *Ridderinkhof et al., 2004*; *Bissonette et al., 2008*; *Owen et al., 1991*; *Dias et al., 1997*). The mPFC interacts with various brain regions to support cognitive functions (*Uddin, 2021*; *Wimmer et al., 2015*; *Aston-Jones and Cohen, 2005*; *Sadacca et al., 2017*; *Cerpa et al., 2021*), and lesion and pharmacology work has pointed to the importance of the locus coeruleus-norepinephrine (LC-NE) input (*Lapiz and Morilak, 2006*; *Tait et al., 2007*; *Newman et al., 2008*; *McGaughy et al., 2008*). However, the precise cellular and circuit mechanisms underlying LC modulation of the mPFC in the context of set shifting are not well understood.

We trained mice to perform a set-shifting task, where they learned to switch attention to discriminate complex sensory cues. Inhibiting genetically defined LC-NE neurons or their projections to the mPFC severely impaired switching behavior, highlighting the importance of the LC-mPFC circuit. Next, to reveal the neurophysiological substrates, we combined chemogenetic inhibition of the LC with calcium imaging of the mPFC in task-performing mice. We discovered that mPFC prominently responded to attentional switching from single cell to population levels, and that LC inhibition dramatically affected mPFC processing across several domains: (1) a greater proportion of mPFC neurons became responsive to switching-related variables; (2) the tuning of individual neurons was broadened; (3) population dynamics associated with attentional switching was disrupted; and (4) population encoding of switching was impaired. Together, our data provide new cellular and circuit-level insights into LC-NE modulation of mPFC activity that support attentional switching.

## Results

We trained mice to perform the freely-moving attentional set-shifting task (*Bissonette et al., 2008*; *Birrell and Brown, 2000*; *Garner et al., 2006*; *Colacicco et al., 2002*) based on procedures described in previous studies (*Liston et al., 2006*; *Snyder et al., 2012*) (Methods). In brief, mice learned to discriminate complex sensory cues by associating a relevant stimulus feature to reward (*Figure 1a and b*). In most stages of the task (simple discrimination, compound discrimination, intra-dimensional reversal, intra-dimensional shift), the rules were different but the relevant stimulus remained in the perceptual dimension of digging medium. In the stage of extra-dimensional shift, the relevant stimulus shifted to the dimension of odor. Mice learned each rule change in a single session, but typically took more trials to complete extra-dimensional shift (e.g. trials to reach criterion: intra-dimensional reversal (IDS) vs. extra-dimensional shift (EDS), 10±1 trials vs. 17±1 trials, $p<0.001$, *Figure 1c*; *Birrell and Brown, 2000*; *Snyder et al., 2012*; *Bissonette et al., 2013*; *Lapiz et al., 2007*). According to learning theories, the improved performance in intra-dimensional reversal (fewer trials to reach performance criterion when all cues are novel but the relevant stimulus feature remains in the same perceptual dimension as previous rules) strongly suggests that mice attend to the perceptual dimension of digging medium while ignoring the perceptual dimension of odor, and that solving extra-dimensional shift involves a switch in the attended perceptual dimension (attentional switching), rather than purely responding to specific exemplar cues (*Roberts et al., 1988*; *Mackintosh, 1975*). Our current work is focused on revealing the neural substrates underlying such attentional switching across perceptual dimensions.

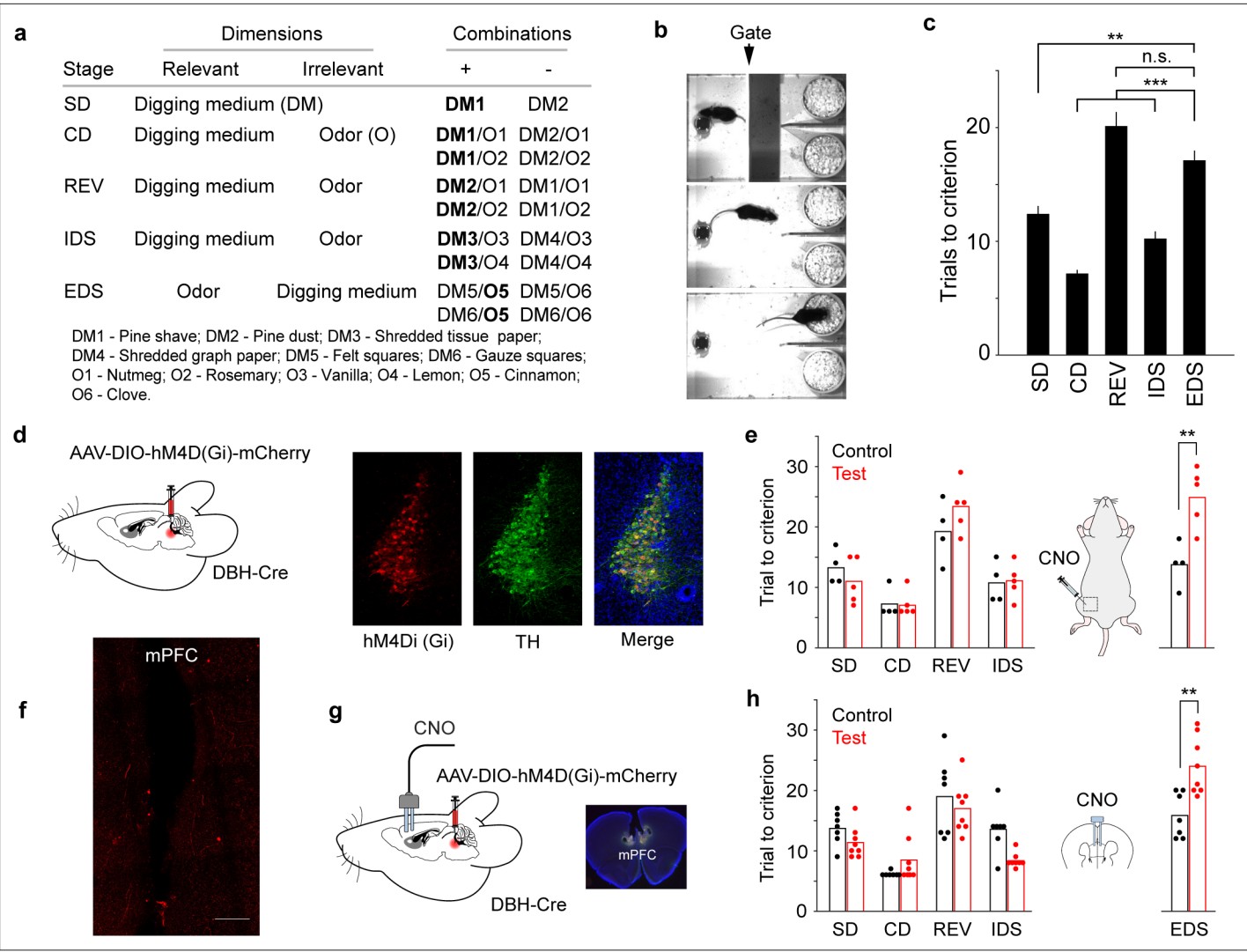

**Figure 1.** Inhibiting locus coeruleus-norepinephrine (LC-NE) neurons or their terminals in the medial PFC (mPFC) impairs switching behavior. (**a**) Task overview. (**b**) Example frames (top to bottom) showing when the mouse was in the waiting area, approaching the bowls, and digging. (**c**) Task performance (total number of trials to criterion) varied across stages: SD – simple discrimination, 12±1 trials; CD – compound discrimination, 7±1 trials; REV – intra-dimensional reversal, 20±1 trials; IDS – intra-dimensional shift, 10±1 trials; EDS – extra-dimensional shift, 17±1 trials. Repeated-measure ANOVA, F(4, 92)=47.8, $p<0.001$, n=24. Post hoc Tukey-Kramer tests: EDS vs. SD, $p=0.003$; EDS vs. CD, $p<0.001$; EDS vs. REV, $p=0.20$; EDS vs. IDS, $p<0.001$; SD vs. CD, $p<0.001$; SD vs. REV, $p<0.001$; SD vs. IDS, $p=0.18$; CD vs. REV, $p<0.001$; CD vs. IDS, $p<0.001$; REV vs. IDS, $p<0.001$. Note that in (**c**) statistical significance was only indicated when comparing EDS to other stages. (**d**) Schematic of DREADD inhibition in the LC and histological images showing DREADD(Gi) and TH (Tyrosine Hydroxylase) expression in the LC of a DBH-Cre mouse. (**e**) Task performance in the control (n=4, WT) and test (n=5) groups. Following systemic CNO injections, test group mice took more trials to complete extra-dimensional shift (EDS). Trials to reach performance criterion: control vs. test, 14±2 trials vs. 25±2 trials, $p=0.007$, t(7) = −3.8. (**f**) Histology showing terminal expression of mCherry in the mPFC. Scalebars: 100 µm. (**g**) Schematic of inhibiting LC terminals in the mPFC and histology displaying cannula placement in the mPFC. (**h**) Task performance in the control (n=7, WT) and test (n=8) groups. Following localized Clozapine N-oxide (CNO) injection, test group mice took more trials to complete EDS (Trials to reach criterion, control vs test: 16±1 trials vs. 24±2 trials, $p=0.003$, t(13) = −3.6).

The online version of this article includes the following source data and figure supplement(s) for figure 1:

**Source data 1.** Data to generate c, e, h.

**Figure supplement 1.** Mice learned to switch attention and LC inhibition impaired this process.

## Inhibiting LC-NE neurons or their input to the mPFC impairs switching behavior

First, we wanted to determine whether the LC-mPFC circuit was required for attentional switching. Previous studies suggested the importance of this circuit by lesioning ascending NE fibers or local

pharmacology in the mPFC (*Lapiz and Morilak, 2006*; *Tait et al., 2007*; *McGaughy et al., 2008*; *Lapiz et al., 2007*; *Arnsten et al., 2012*), which broadly targeted NE signaling. To selectively target and perturb genetically-defined LC-NE neurons, we employed a transgenic approach to conditionally express the Cre-dependent inhibitory DREADD receptor hM4Di in the LC of DBH-Cre mice (Test group, *Figure 1d*). Dopamine Beta Hydroxylase (DBH) is a key enzyme for NE synthesis and downstream of dopamine. Thus, DBH serves as a specific marker for NE-synthesizing neurons. Control group mice were DBH- littermates and received Clozapine N-oxide (CNO) administrations the same way as test group (immediately after IDS and 60 min prior to EDS). DREADD inhibition of LC-NE neurons impaired switching behavior in EDS as test group mice took more trials to reach performance criterion (*Figure 1e*, trials to reach performance criterion: control vs. test, 14±2 trials vs. 25±2 trials, *p*=0.007). Similar behavioral effects were observed when a second control group mice were DBH-Cre expressing hM4Di but received saline injections (*Figure 1—figure supplement 1a*). Together, these data strongly implicate that the behavioral impairment is specific to LC inhibition, instead of nonspecific effects of genetic background, viral expression, or DREADD agonist.

LC-NE neurons innervate the mPFC, but the specific role of their direct input has not been fully explored. To address this question, we expressed hM4Di in the LC (as in *Figure 1d*) and locally infused CNO in the mPFC via bilateral cannula implants to perturb the terminals of LC neurons (*Figure 1f and g*, as in *Liu et al., 2021*; *Mahler et al., 2014*). This approach allows for targeted and specific perturbation of LC input to the mPFC. Control group mice (DBH-) also had cannula implants and received CNO injections in the same manner. Suppressing LC-NE terminals in the mPFC also induced pronounced behavioral deficits in EDS, with test group mice requiring more trials to switch to the new perceptual dimension (*Figure 1h*, trials to reach the criterion, control vs. test: 16±1 trials vs. 24±2 trials, *p*=0.003). Importantly, EDS performance across different control groups was not different (control group in *Figure 1c* vs. *Figure 1e*, *p*=0.14; control group in *Figure 1c* vs. *Figure 1h*, *p*=0.47). The behavioral impairment of DREADD inhibition was also robust against different control groups (e.g. LC inhibition

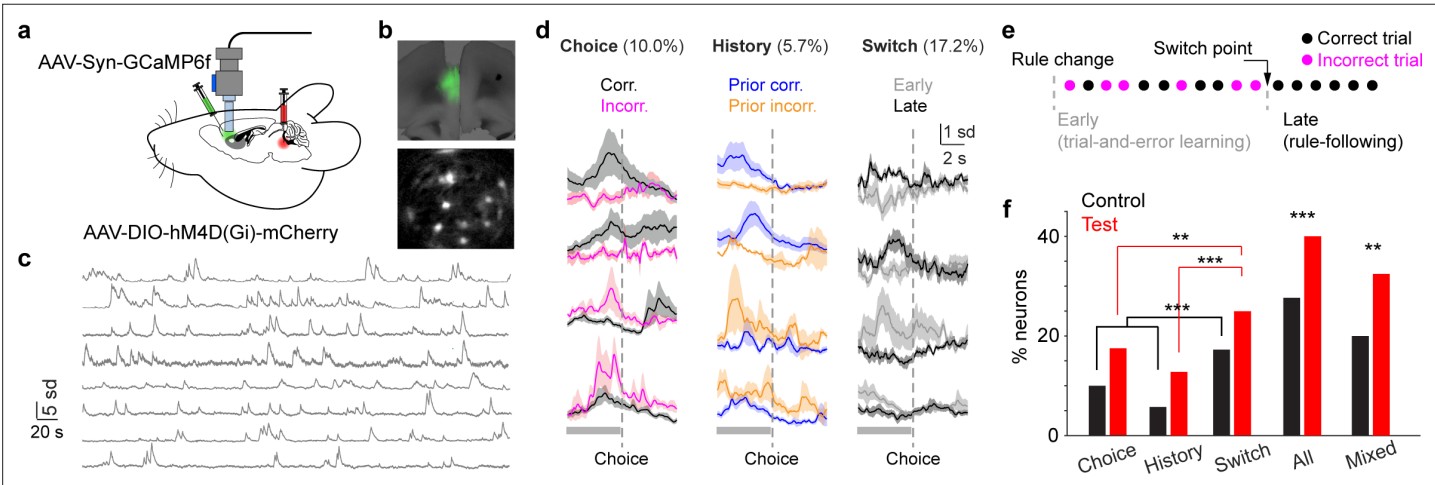

**Figure 2.** Locus coeruleus (LC) inhibition enhances medial PFC (mPFC) engagement and broadens tuning. (**a**) Illustration of miniscope recording in the mPFC with DREADD inhibition in the LC. (**b**) Top: Histology showing lens implant and GCaMP6f expression in the mPFC (prelimbic). Bottom: Snapshot of miniscope recording during behavior. (**c**) Example time series of fluorescence signals. Over 50 ROIs were acquired from this session. (**d**) Left to right: Example traces of individual mPFC neurons responding to choice (left), trial history (middle), and switch (right) based on activity prior to choice (gray bars). (**e**) Example behavioral progression. Each dot represents a trial. We define the initial mixed correct and incorrect trials (rule-learning) and the last set of consecutive correct trials (rule-following) as two different states in switching behavior. (**f**) Bar plots showing the percentage of mPFC neurons responding to task-related variables in the control (black) and test (red) groups. Control vs. test, choice responsive: 10% (59/593) vs 17% (72/432), *p*=0.002, $\chi^2$=10.1; history responsive: 6% (34/593) vs 13% (55/432), *p*<0.001, $\chi^2$=15.4; switch responsive: 17% (102/593) vs 25% (106/432), *p*=0.004, $\chi^2$=8.3; overall fraction of responsive neurons: 27% (159/593) vs 40% (172/432), *p*<0.001, $\chi^2$=19.3; the fraction of mixed tuning neurons among all responsive neurons: 20% (31/159) vs 32% (55/172), *p*=0.01, $\chi^2$=6.7, Chi-squared test.

The online version of this article includes the following figure supplement(s) for figure 2:

**Figure supplement 1.** The effect of LC inhibition on calcium activity of mPFC neurons.

**Figure supplement 2.** Mixed tuning in the mPFC.

**Figure supplement 3.** Dual-camera tracking of mouse behavior.

group in *Figure 1e* vs. control group in *Figure 1c*, *p*<0.001; terminal inhibition group in *Figure 1h* vs. control group in *Figure 1c*, *p*<0.001). Our results add further to recent gain-of-function work (*Cope et al., 2019*), providing compelling evidence for the critical involvement of the LC-mPFC circuit in attentional switching.

## LC inhibition enhances mPFC engagement and broadens single-neuron tuning

To assess the neurophysiological effects of LC-NE signaling on mPFC activity, we simultaneously induced the expression of Gi-DREADD in the LC and GCaMP6f in the mPFC (*Figure 2a–c*). We monitored mPFC activity while inhibiting LC-NE neurons in task-performing mice (432 neurons from four test mice, 89.6±4.0% TH+ neurons in the LC expressing hM4Di). Control mice (DBH-) expressed GCaMP6f in the mPFC and received agonist injections in the same manner (593 neurons from four control mice). DREADD agonist CNO was systemically administered to both control and test group mice immediately after IDS and 60 min prior to EDS, and test mice exhibited similar behavioral impairment (*Figure 1—figure supplement 1b*). We evaluated the overall effect of LC inhibition on calcium activity of mPFC neurons. The frequency and amplitude of calcium transients did not differ between test and control groups, but the test group exhibited a small (~5%) reduction in transient duration (*Figure 2—figure supplement 1*). Importantly, control mice (pooled from *Figure 1e,h*, *Figure 1—figure supplement 1a,b*) took more trials to complete EDS than IDS (Trials to criterion: IDS vs. EDS, 11±1 trials vs. 15±1 trials, *p*=0.006, *Figure 1—figure supplement 1c*). In a separate analysis, we excluded 3 sessions from this pooled control group, where task performance in IDS exceeded 95% threshold (15 trials) inferred from the naïve control group (*Figure 1c*), and the behavioral difference was robust (Trials to criterion: IDS vs. EDS, 11±1 trials vs. 15±1 trials, *p*=0.003, *Figure 1—figure supplement 1d*). These results further support the validity of attentional switching in these mice (as in *Figure 1c*).

Next, we examined how single-neuron response during attentional switching was affected by LC inhibition. Following recent work (*Cho et al., 2020*; *Cho et al., 2023*), we used the time of choice (digging) to infer decision formation and classified the representation of individual mPFC neurons based on their pre-choice activity. We first presented the results from the control group. We identified subgroups of mPFC neurons whose activity was tuned to different task-related variables, such as choice, trial history, and the putative switch of attention (*Figure 2d*, Methods), consistent with a series of previous work (e.g. *Mansouri et al., 2006*; *Norman et al., 2021*; *Spellman et al., 2021*; *Lui et al., 2021*; *Jercog et al., 2021*; *Del Arco et al., 2017*; *Durstewitz et al., 2010*). Since the rule change was not cued, at the beginning of extra-dimensional shift, animals followed the previous rule and attended to the perceptual dimension of digging medium and ignored odor cues (*Birrell and Brown, 2000*). Through trial-and-error learning animals eventually switched their attention to the perceptual dimension of odor (*Mansouri et al., 2006*; *Spellman et al., 2021*). Following prior studies (*Sleezer et al., 2017*; *Sleezer et al., 2016*), we inferred the early mixed correct and incorrect trials and the late set of consecutive correct trials as different states of switching behavior (Early: trial-and-error learning; Late: rule following. *Figure 2e*). Notably, more neurons responded to switch than to choice or trial history (fraction of neurons, switch vs. choice: 17% (102/593) vs 10% (59/593), *p*<0.001; switch vs. history: 17% (102/593) vs 6% (34/593), *p*<0.001, chi-squared test. *Figure 2f*), suggesting the importance of representing this task-related variable in the mPFC. We further noted that a considerable fraction of mPFC neurons responded to more than one task-related variable (mixed tuning *Mansouri et al., 2006*; *Lui et al., 2021*; *Rigotti et al., 2013*; *Kim et al., 2016*; *Pinto and Dan, 2015*, *Figure 2—figure supplement 2*. Data from individual mice were in *Supplementary file 1*). To better determine choice-related behavior, a second side-view camera was set up, and the temporal difference between digging onset estimated from the two cameras was minimal (*Figure 2—figure supplement 3*), confirming the fidelity of the timestamps used for data analysis.

During LC inhibition, we also observed more mPFC neurons tuned to switch (fraction of neurons in the test group, switch vs. choice: 25% (106/432) vs 17% (72/432), *p*=0.004; switch vs history: 25% (106/432) vs 13% (55/432), *p*<0.001, chi-squared test). More importantly, in comparison to the control group, LC inhibition engaged a greater fraction of mPFC neurons responding to task-related events (*Figure 2f*, control vs. test, choice responsive: 10% (59/593) vs 17% (72/432), *p*=0.002; history responsive: 6% (34/593) vs 13% (55/432), *p*<0.001; switch responsive: 17% (102/593) vs 25% (106/432),

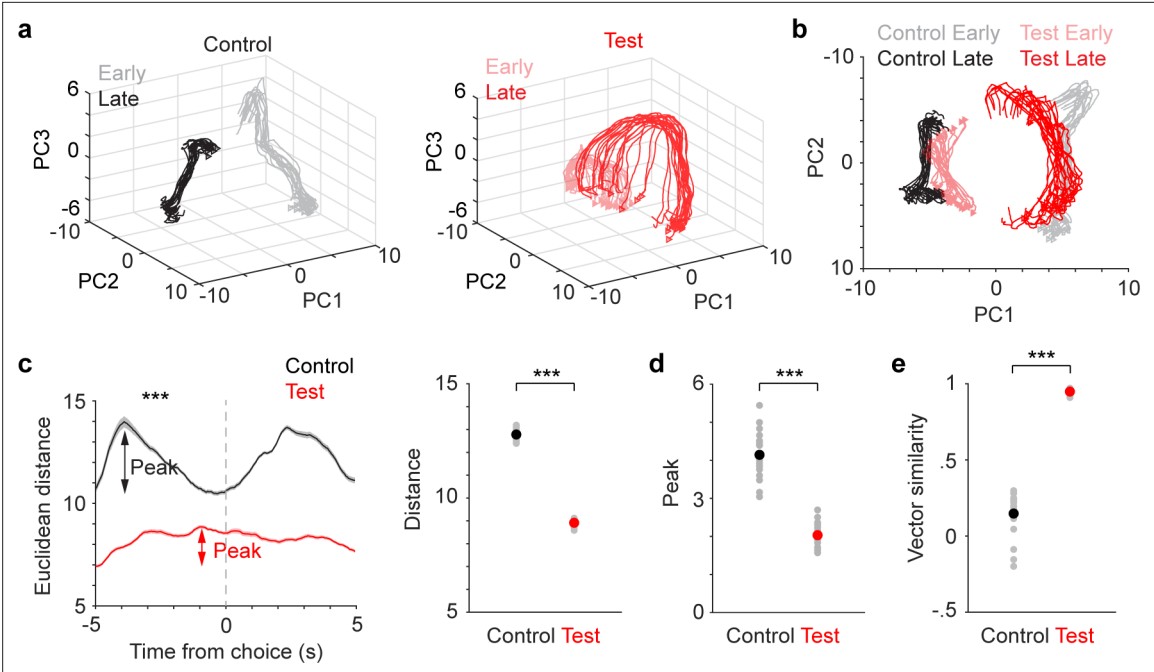

**Figure 3.** Locus coeruleus (LC) inhibition dampens medial PFC (mPFC) population dynamics during switching. (**a**) Population vectors of mPFC activity representing early (light color) and late (dark color) states in control (black, left) and test (red, right) groups. Each line represents a population vector from a subset of neurons. (**b**) Projection of population vectors in (**a**) onto the first two PCs. (**c**) Left: Euclidean distance (mean ± SEM) between state vectors aligned to choice for control (black) and test (red) groups. Arrows indicate maximal fluctuations prior to choice (peak). Right: Comparison of Euclidean distance quantified prior to choice for control (black) and test (red) groups. Control vs. test, 12.9±0.08 vs 8.9±0.05, *p*<0.001, rank sum = 610, n=20. Sample size represents number of bootstraps. (**d**) Comparison of peak Euclidean distance quantified prior to choice for control (black) and test (red) groups. Control vs. test: 4.0±0.10 vs 2.1±0.06, *p*<0.001, rank sum = 610, n=20. (**e**) Comparison of vector similarities between the early and late states for control and test groups. Correlation coefficient, control vs. test: 0.15±0.04 vs 0.95±0.01, *p*<0.001, rank sum = 210, n=20. Black and red dots indicate group mean in (**c–e**).

The online version of this article includes the following source data and figure supplement(s) for figure 3:

**Source data 1.** Data to generate c–e.

**Figure supplement 1.** State vectors were more similar with LC inhibition.

*p*=0.004; overall fraction of responsive neurons: 27% (159/593) vs 40% (172/432), *p*<0.001, chi-squared test). The fraction of mixed-tuning neurons was also enhanced with LC inhibition (20% (31/159) vs 32% (55/172), *p*=0.01). Our results show that LC inhibition increases mPFC engagement in the task and broadens the tuning of individual neurons.

## LC inhibition impedes dynamic changes in population activity during switching

Our single-neuron analysis suggests the importance of encoding attentional switch in the mPFC as more neurons were tuned to this parameter in both control and test groups (*Figure 2f*, *Supplementary file 1*). Neuronal ensembles have been proposed to be the functional unit of the nervous system (*Yuste, 2015*; *Pouget et al., 2000*; *Ebitz and Hayden, 2021*; *Saxena and Cunningham, 2019*). They can better represent information than single neurons (*Jercog et al., 2021*; *Meyers et al., 2008*; *Driscoll et al., 2017*), especially in higher-order association areas where single neurons exhibit mixed tuning (*Rigotti et al., 2013*; *Fusi et al., 2016*; *Tye et al., 2024*). Thus, we sought to determine whether and how attentional switching was represented at the population level. We first employed a dimensionality-reduction approach to assess mPFC population dynamics (Methods). Specifically, we examined whether mPFC dynamic processes represent the putative switch of attention. Principle component analysis (PCA) was applied to population activity of mPFC neurons around the time of choice in the early and late switching states (*Figure 2e*), and the degree of separation between the resulting low-dimensional state vectors was quantified. In the control group mice, we identified an

overall prominent separation between the two population vectors representing early and late states (*Figure 3a and b*, gray vs. black traces), strongly suggesting that a shift in population dynamics is associated with attentional shifting across perceptual dimensions. The vector separation also exhibited transient fluctuations prior to choice, first increasing and then decreasing (*Figure 3c*, black), suggesting a dynamic decision-related population encoding process underlying the behavioral transitions. Overall, our results suggest that mPFC dynamics reflect the changes in switching behavior and learning of the new rule.

How would LC-NE input affect mPFC dynamics during switching? In test group mice, the same dimensionality-reduction analysis revealed that the low-dimensional population state vectors (early and late) were less separable (*Figure 3a and b*, light and dark red traces), and the distance between the two state vectors was greatly reduced compared to the control group (*Figure 3c*, control vs. test, 12.9±0.08 vs 8.9±0.05, *p*<0.001). In addition, LC inhibition prominently dampened the pre-choice dynamic fluctuations (*Figure 3c and d*. Peak, control vs. test: 4.0±0.10 vs 2.1±0.06, *p*<0.001). LC inhibition also rendered the population state vectors more similar to one another (*Figure 3*, *Figure 3—figure supplement 1*) (Correlation coefficient between early and late state vectors, control vs. test: 0.15±0.04 vs 0.95±0.01, *p*<0.001). Together, our results show that LC inhibition dampens and impedes mPFC dynamics during switching.

## LC inhibition impairs population encoding of switching

To gain further insights into mPFC representation of attentional switching and the effects of LC inhibition, we turned to the hidden Markov model (HMM), which has been successfully implemented to link neuronal activity patterns to animal behavior (e.g. *Durstewitz et al., 2010*; *Bagi et al., 2022*;

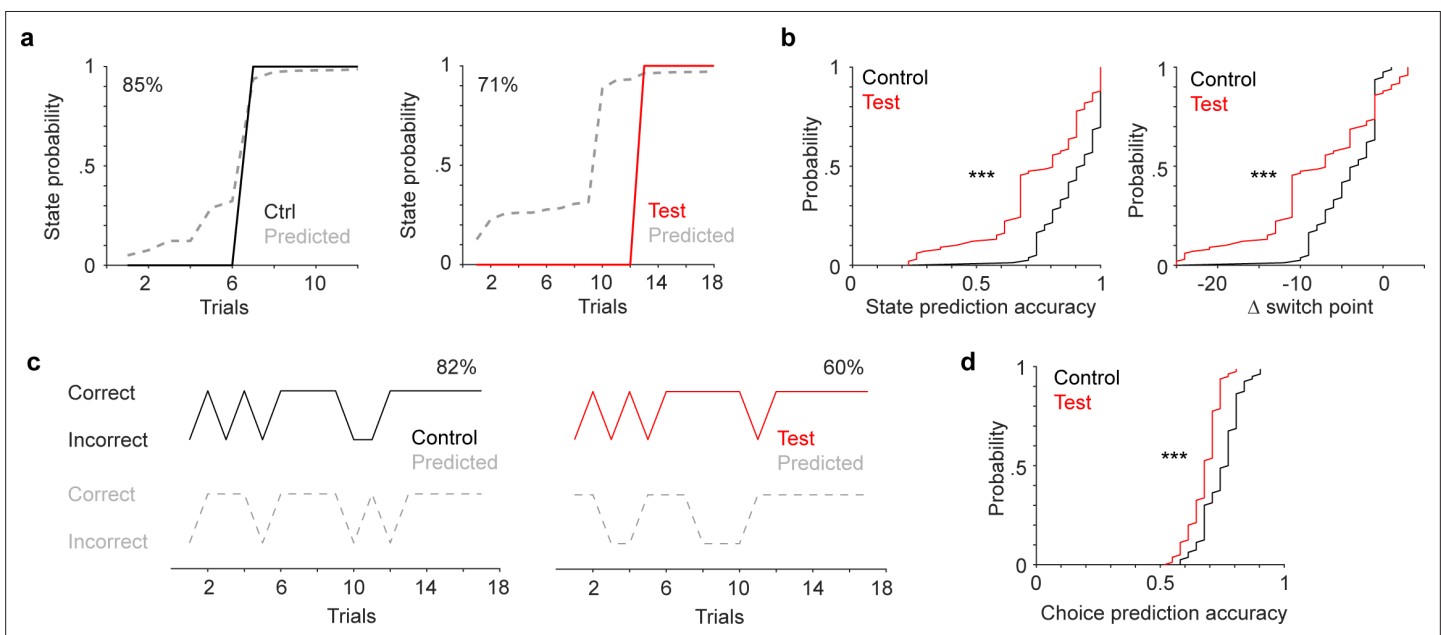

**Figure 4.** Locus coeruleus (LC) inhibition impairs medial PFC (mPFC) encoding capacity of switching. (**a**) Example behavioral state progression (solid curve: 0-early, 1-late) and hidden Markov model (HMM) predicted state progression (dashed curve) in a control session (black, left) and a test session (red, right). State prediction accuracy is 85% (control) and 71% (test). (**b**) Left: Cumulative distribution of the accuracy of predicting behavioral states in control (black) and test (red) groups. Sample size represents the total number of iterations that the model was tested (20 times per recording, four control mice, and four test mice). Control vs. test: 0.89±0.01 vs 0.75±0.02, *p*<0.001, rank sum = 7,828. Right: Cumulative distribution of the accuracy of predicting switch point in control (black) and test (red) groups. Control vs. test: −4±1 trials vs −8±1 trials, *p*<0.001, rank sum = 7,398. (**c**) Example sequences of animals' choices (solid, top) and generalized linear model (GLM) predicted choices (dashed, bottom) in a control session (black, left) and a test session (red, right). Prediction accuracy is 82% (control) and 60% (test). (**d**) Cumulative distribution of the accuracy of predicting trial-by-trial choices in control (black) and test (red) groups. Control vs. test: 0.75±0.01 vs 0.68±0.01, *p*<0.001, rank sum = 4,980.

The online version of this article includes the following source data and figure supplement(s) for figure 4:

**Source data 1.** Data to generate b, d.

**Figure supplement 1.** LC inhibition did not affect gross motor functions.

*Recanatesi et al., 2022*; *Mazzucato et al., 2015*). In brief, we assume that population activity vectors, represented as calcium signals from simultaneously recorded neurons, are adjacent to one another in the neuronal state space when the same behavior is executed. We clustered mPFC population vectors (5 s prior to choice) into a set of discrete states (hidden states), and assessed the relationship between these neuronal states and the observed behavioral patterns/states of the mice. Specifically, the behavioral states are the early rule learning state and the late rule-following state (as depicted in *Figure 2e*). We identified low-dimensional factors from the high-dimensional population vectors, and fitted HMM to these factors to infer the hidden state of each trial (Methods). We quantified model performance by comparing each trial's neuronal state to the behavioral state with two measures: (1) the overall accuracy of predicting the state of individual trials (early or late) in each session; and (2) the accuracy of predicting where state transition occurred (the onset of late state: switch point, *Figure 2e*). Both measures showed that model prediction was less accurate when LC was inhibited (*Figure 4a and b*, state prediction accuracy, control vs. test: 89±1% vs 75 ± 2%, *p*<0.001; Δ switch point, control vs. test: –4±1 trials vs. –8±1 trials, *p*<0.001).

We further assessed whether mPFC activity can track animals' choices on each trial (correct vs. incorrect). We applied a generalized linear model (GLM) to predict the upcoming choice on individual trials. Specifically, we included the first three principle components of the 5 s pre-choice population activity as regressors (Methods). We discovered that LC inhibition reduced the accuracy of trial-by-trial choice predictions (*Figure 4c and d*, control vs. test: 75±1% vs 68±1%, *p*<0.001). Video analysis found no significant difference in overall locomotion or reaction time between test and control group mice (*Figure 4—figure supplement 1*), strongly implicating that the observed behavioral and neurophysiological effects were not due to apparent changes in motivation or motor functions. Together, our data show that LC inhibition produces a marked deficiency in mPFC population encoding of attentional switching processes, suggesting that impaired mPFC dynamics and encoding capacity underlie the behavioral deficits.

## Discussion

Our current work is aimed to uncover the neurophysiological substrates underlying attentional switching (set shifting) processes. We trained mice to switch attention to discriminate complex stimulus features comprising perceptual dimensions of digging medium and odor. Inhibiting genetically defined LC-NE neurons or their projections to the mPFC similarly impaired switching behavior, highlighting the importance of the LC-mPFC circuit. To reveal the neurophysiological substrates, we combined chemogenetic inhibition of the LC with calcium imaging of the mPFC in task-performing mice. We discovered that the putative switch of attention was prominently represented in the mPFC, and LC inhibition dramatically altered mPFC activity from single-cell to population levels. A greater proportion of mPFC neurons became responsive to task-related variables, and the tuning of these neurons was broadened. Furthermore, LC inhibition disrupted mPFC population dynamics and impaired the encoding capacity of switching. Together, our data provide new cellular and circuit-level insights into LC modulation of mPFC activity during attentional switching.

Our analysis revealed that LC inhibition enhanced the engagement of mPFC neurons in the task. This observation may appear counterintuitive at first glance, but amplification of neuronal responses in a region could transmit noisy information to downstream circuits (*Lam et al., 2022*; *Selimbeyoglu et al., 2017*; *Wang et al., 2010*), impairing brain functions. Furthermore, the link between broadened tuning and impaired switching behavior is reminiscent of the relationship between the changes in tuning properties of sensory neurons and perceptual behavior (e.g. *Poort et al., 2022*; *Poort et al., 2015*; *He et al., 2017*; *Goel et al., 2018*). It is plausible that similar to sensory areas, an appropriate level of mixed tuning in association areas is optimal for population coding of cognitive processes (*Tye et al., 2024*), and that too broad tuning would deteriorate population representations of task- and decision-related features. This prediction needs to be tested in future computational work.

Given that inhibiting LC-NE terminals impaired switching behavior in a similar manner as inhibiting LC-NE neurons (*Figure 1e and h*), we interpret the observed neurophysiological effects in the mPFC during LC inhibition (*Figures 2–4*) as at least partially mediated by the direct LC input. NE exerts both excitatory and inhibitory influences through distinct types of adrenergic receptors expressed in different cell types (*Berridge and Waterhouse, 2003*). By preferentially binding to specific types of adrenergic receptors in a concentration-dependent way, NE is proposed to mediate downstream

neuronal activity and behavior in a non-linear manner (*Aston-Jones and Cohen, 2005*). Interestingly, a recent study in the orbital prefrontal cortex showed that the reduction of NE and downregulation of alpha-1 receptors led to decreased activity in GABAergic interneurons (*Li et al., 2023*). In addition, prominent gamma synchrony between bilateral mPFC was important to support set shifting and population dynamics (*Cho et al., 2020*; *Cho et al., 2023*). It is thus plausible that the lack of LC-NE input diminishes the engagement of GABAergic interneurons in the mPFC, leading to elevated noisy neuronal activity, broadened tuning, and reduced population representations. However, it is important to note that LC-NE neurons can co-release other neurotransmitters, such as dopamine and neuropeptides (*Berridge and Waterhouse, 2003*; *Takeuchi et al., 2016*; *Kempadoo et al., 2016*). In the absence of further control experiments to confirm the suppression of NE release, the observed effects on mPFC may or may not be directly due to NE. Future studies are needed to better delineate the involvement of specific neurotransmitters, cell types, and receptors in flexible decision making.

Our analysis suggests that attentional switching was prominently represented at both single-cell and population levels in the mPFC, and that LC inhibition led to pronounced changes in neuronal coding and population dynamics. Abrupt network transitions have been observed in the mPFC of rats performing set-shifting or probabilistic alternative choice task (*Durstewitz et al., 2010*; *Karlsson et al., 2012*). Disrupted mPFC encoding or population activity patterns were reported when perturbing thalamic drive or callosal PV projections in mice (*Cho et al., 2023*; *Benoit et al., 2022*). Together, these findings underscore a key insight: while many brain circuits can influence mPFC function, their effects may converge onto a small set of general operational principles, such as modulating the tuning properties of individual neurons and/or orchestrating ensemble dynamic transitions during complex cognitive processes. Identifying these principles is vital for advancing our understanding of how prefrontal cortex contributes to higher-order cognition and how its functions can be affected in various contexts.

In both intra-dimensional shift and extra-dimensional shift, all cues are novel but the rules differ. Learning theories posit that improved performance in intra-dimensional shift (fewer trials to reach performance criterion when all cues are novel but the relevant stimulus feature remains in the same perceptual dimension, e.g. digging medium in our task) is due to subject's ability to readily attend to the superordinate features of sensory cues (perceptual dimensions - digging medium vs. odor), and that solving the extra-dimensional shift rule requires a switch in the attended perceptual dimension, rather than merely responding to individual novel cues (*Roberts et al., 1988*; *Mackintosh, 1975*). Thus, the behavioral changes observed in extra-dimensional shift (more trials to reach performance criterion) reflect the adaptive processes underlying the reallocation of attention, instead of novelty response. Based on this understanding, our current work builds on a longstanding tradition in the field that uses a single extra-dimensional shift to test attentional switching (e.g. *Dias et al., 1996b*; *Bissonette et al., 2008*; *Lapiz and Morilak, 2006*; *McGaughy et al., 2008*; *Birrell and Brown, 2000*). In this context, animals are naïve to the rule change and would solve the problem 'on the fly,' without relying on prior learning or knowledge. Our findings shed new light on how the LC-mPFC circuit supports such *de novo* attentional switching processes. Furthermore, limited evidence suggests that solving the switching problem 'on the fly' (initial encounter) or based on experience/internal models (repeated testing) involves different mechanisms (*Dias et al., 1997*). A comprehensive comparison between these settings could provide valuable insights and further advance our understanding of cognitive flexibility.

Another limitation in the current study is that neurophysiological analyses were entirely from EDS. Without comparing with other task stages (e.g. REV, IDS), it is uncertain to what extent the observed neuronal changes are specific to EDS. Future experiments should examine the behavioral and neurophysiological effects with LC inhibition to determine the specificity of LC-NE modulation of the mPFC during attentional switching.

Our work contributes to the growing interest in revealing neural mechanisms underlying more natural, ethologically relevant behavior (*Dennis et al., 2021*; *Parker et al., 2020*). Admittedly, such behavioral paradigms may not afford the level of task control more commonly seen in restrained, operant paradigms. Nevertheless, the challenge of dissociating movement-related signal from sensory- or decision-related signal is present not only in freely-moving, but also restrained settings (*Zagha et al., 2022*; *Musall et al., 2019*; *Steinmetz et al., 2019*; *Stringer et al., 2019*). Comprehensive behavioral tracking and motif analysis (e.g. *Wiltschko et al., 2015*; *Markowitz et al., 2023*)

will help to identify whether specific behavioral patterns are associated with attentional switching behavior. Ultimately, cognitive processes are not independent from sensory or motor processes. Cognition, perception, and action may be implemented in a distributed rather than isolated manner (*Parker et al., 2020*; *Zagha et al., 2022*; *Cisek and Kalaska, 2010*).

# Methods

**Key resources table**

| Reagent type (species) or resource | Designation | Source or reference | Identifiers | Additional information |
|---|---|---|---|---|
| Genetic reagent (*Mus musculus*) | DBH-Cre | MMRRC | 036778-UCD | |
| Strain, strain background (*M. musculus*) | C57BL/6 J | JAX | Strain #:000664 | |
| Recombinant DNA reagent | AAV8-hSyn-DIO-hM4D(Gi)-mCherry | Addgene | 44362-AAV8 | |
| Recombinant DNA reagent | AAV1-Syn-GCaMP6f-WPRE-SV40 | Addgene | 100837-AAV1 | |
| Antibody | Rabbit polyclonal anti-Tyrosine Hydroxylase | Thermo Fisher | OPA1-04050 | Primary 1:1000 |
| Antibody | Goat anti-rabbit polyclonal secondary | Thermo Fisher | A32731/A32740 | Secondary 1:500 |
| Software, algorithm | Matlab | MathWorks | RRID:SCR_001622 | |
| Other | Miniscope | Inscopix | nVista | |
| Chemical compound | Clozapine-N-oxide | Enzo Life Sciences | BML-NS105-0005 | |

## Surgery

All experimental procedures were approved by the UC Riverside Animal Care and Use Committee (AUP20220030). Mice of mixed sex, aged 8–16 weeks were included in the study. Mice were C57BL/6 J and DBH-Cre (B6.FVB(Cg)-Tg(Dbh-cre)KH212Gsat/Mmucd, 036778-UCD, MMRRC), singly housed in a vivarium with a reversed light-dark cycle (9a-9p). All surgical procedures were conducted under aseptic conditions, maintaining body temperature with a heating pad. Anesthesia was induced using a mixture of isoflurane (2–3%), and mice were positioned within a precise digital small-animal stereotaxic apparatus (Kopf Instruments and RWD). Before surgery, hair was gently removed from the dorsal head area, ophthalmic ointment was applied to protect the eyes, and the incision site was sanitized with betadine. All measurements were referenced to bregma for virus/implant surgeries. Viral injections were accomplished using a microinjection needle coupled with a 10 µl microsyringe (WPI). The virus was delivered at a controlled rate of 0.03 µl/min via a microsyringe pump (WPI). Following the completion of the injection, a 10 min interval was allowed before slowly withdrawing the needle. Subsequent to viral infusions, nylon sutures were used to close the incision. Animals received 0.10 mg/kg buprenorphine and 0.22% enrofloxacin and were placed in their respective home cages over a heating pad at 37 °C. After full recovery from anesthesia, the subjects were returned to vivarium.

For LC inhibition, dopamine-β-hydroxylase (DBH)-Cre mice received bilateral injections of AAV carrying Gi-DREADD receptors (AAV8-hSyn-DIO-hM4D(Gi)-mCherry) into the LC (AP: –5.1, ML: 0.95, DV: –3.5, and –3.7 mm, 0.3 µl each depth). For calcium imaging, AAV containing GCaMP6f (AAV1-Syn-GCaMP6f-WPRE-SV40) was injected into the medial prefrontal cortex (Prelimbic; AP: 1.8, ML: 0.3, DV: –2.0, and –2.4 mm, 0.3 µl). Following injections, a 30-gauge needle was inserted to create space and reduce tissue resistance to facilitate lens insertion; however, no tissue was aspirated. A gradient refractive index lens (GRIN lens, Inscopix) with a diameter of 0.5 mm and approximately 6 mm in length was gradually lowered through the craniotomy, allowing tissue decompression. This lens was positioned to target AP: –1.8, ML: 0.3, DV: 2.2 mm. The same coordinates were applied for the cannula implantation used in LC terminal inhibition. Lens implants were securely attached to the skull using a layer of adhesive cement (C&B Metabond, Parkell), followed by dental cement (Ortho-Jet, Lang Dental). To protect the lens, a layer of silicone rubber was applied as a protective cover.

Following virus incubation, mice were once again anesthetized under isoflurane and securely positioned in the stereotaxic setup. Baseplates were affixed around the GRIN lens to provide structural support for the attachment of the miniaturized microscope. The top surface of the exposed GRIN lens was meticulously cleaned using a cotton-tipped applicator dipped in a solution of 15% isopropyl

alcohol diluted with ddH$^2$O. A miniaturized microscope, equipped with a single-channel epifluorescence and a 475 nm blue LED (Inscopix), was then carefully positioned over the implanted GRIN lens. Adjustments were made along the dorsal-ventral axis to achieve the optimal focal plane for imaging. Subsequently, the microscope/baseplate assembly was secured in place using adhesive cement. The microscope was detached from the baseplates, and a final layer of dental cement was applied to prevent light leakage. A protective plate was placed over the implant until imaging sessions. Mice were singly housed after lens implant.

## Behavior and data acquisition

To assess flexible decision-making in freely moving mice, we adopted the 5-stage testing paradigm of the attentional set-shifting task (AST) (*Liston et al., 2006*; *Snyder et al., 2012*). Two weeks before training, mice were food restricted (85% of initial weight) and handled by the experimenter for 5–7 days. Next, mice were acclimated to the behavioral box (25×40 cm) and experimental setup for 3–4 days, followed by a brief training session to stimulate the innate burrowing/digging behavior to retrieve food reward from the ramekins. Two ramekins were placed at two corners of the behavioral box, both containing 25 mg of cheerios. Throughout the training session, the reward was gradually buried in clean home cage bedding. In each trial, mice were allowed 3–4 min to explore. Mice were considered well-trained once they could consistently dig and retrieve the reward from both locations for 15–20 trials.

The AST consisted of the following stages: (1) simple discrimination (SD), in which animals choose between two digging mediums associated with distinct textures (first dimension), only one of the two stimuli predicts food reward; (2) compound discrimination (CD), in which a second stimulus dimension (two odor cues) is explicitly introduced. Each odor cue is randomly paired with a digging medium in every trial, but the reward is still predicted as in SD; (3) intra-dimensional reversal (REV), which preserves the task-relevant dimension (digging medium) but swaps cue contingencies; (4) intra-dimensional shift (IDS), which preserves the task-relevant dimension (digging medium), but replaces all four cues with novel ones (a new digging medium predicts reward); (5) extra-dimensional shift (EDS), which swaps the previous task-relevant and task-irrelevant dimensions with all cues replaced (a new odor cue predicts reward). All stages were performed within a single day, lasting 3–4 hr. In each trial, the ramekin associated with the relevant stimulus contained a retrievable reward. To avoid the possibility that mice used food odor cues to solve the task, the other ramekin contained a non-retrievable reward (trapped under a mesh wire at the bottom). The two ramekins were placed randomly in the two locations every trial. Mice were allowed to complete a trial (dig one ramekin) within 3 min. Once mice started digging, the other ramekin was immediately removed from the behavioral box. To reach the criterion, the animal has to dig in the correct ramekin six times consecutively and correctly rejecting the incorrect ramekin on at least two occasions.

An overhead CCD camera (Basler acA1300-200um) was set to capture behavior at 20 Hz, controlled by Pylon Viewer Software. Video and calcium recordings were synchronized via a common TTL pulse train (Arduino). Behavioral annotations were done manually post hoc. On the recording day, mice were attached to the miniaturized microscope. Grayscale images were collected at 20 frames per second using 0.01 mW/mm$^2$ of excitation light. Snout, head, tail, and ear tracking were measured using Deep-LabCut (*Mathis et al., 2018*). The network was initially trained with 100 uniformly distributed frames from five videos, followed by an additional iteration to rectify outlier detections. The measurements for distance and speed were computed using the head, where the likelihood of accuracy exceeded 95 percent. After the test, animals were allowed to access food and water *ad libitum* for 3 days before to be transcardially perfused. Following dissection, brains were post-fixed for 24 hr at 4 °C in 4% PFA, and sectioned for immunohistochemistry to label TH +neurons. Specifically, brain sections containing the LC were incubated with a rabbit anti-Tyrosine Hydroxylase (TH) primary antibody (Thermo Fisher, Cat# OPA1-04050; 1:1000), followed by incubation with a goat anti-rabbit IgG secondary antibody conjugated to Alexa Fluor 488 or 594 (Thermo Fisher, Cat# A32731 or A32740; 1:500). Sections were mounted using DAPI-containing mounting medium (Vector Laboratories).

## Locus coeruleus inactivation

On the test day, Clozapine-N-oxide (CNO) was freshly prepared for systemic or local infusions. For systemic injection, we used a concentration of 0.03 mg/kg to minimize potential confound (*Boekhoudt*

*et al., 2017; Gomez et al., 2017; Souza et al., 2022*). CNO was injected immediately after IDS and 60 min before EDS test in both test (Gi-DREADD) and control (DBH-) group mice. Maximal effects of systemic CNO activation were reported to occur after 30 min and last for at least 4–6 hr (*Urban and Roth, 2015; Alexander et al., 2009; Krashes et al., 2013*). A second control group mice (DBH-Cre expressing Gi-DREADD) received saline injections in the same manner. For LC terminal inhibition, we used a CNO concentration of 0.5 mM (*Liang et al., 2020*) diluted in cortex buffer. Mice were bilaterally implanted with stainless steel cannula guide (26 gauge; RWD) targeting the mPFC. Dummy cannulas were used to fill the guide and removed only during the injection period. CNO was infused at a rate of 0.03 µl/min. After infusion, injecting cannulas were left in place for 5 min to allow drug diffusion.

## Image processing

Image processing was executed using Inscopix data processing software (version 1.6), which includes modules for motion correction and signal extraction. Prior to data analysis, raw imaging videos underwent preprocessing, including a 4x spatial down sampling to reduce file size and processing time. No temporal down sampling was performed. The images were then cropped to eliminate post-registration borders and areas where cells were not visible. Prior to the calculation of the dF/F0 traces, lateral movement was corrected. For ROI identification, we used a constrained non-negative matrix factorization algorithm optimized for endoscopic data (CNMF-E) to extract fluorescence traces from ROIs. The detected ROIs were then manually evaluated based on neuronal morphology, ensuring adequate separation from neighboring cells. We identified 128±31 neurons after manual selection, depending on recording quality and field of view (number of identified neurons, control: 153, 260, 156, 24; test: 74, 240, 48, 70).

## Single cell analysis

Calcium signals for each ROI were z-scored and aligned to behavioral events (i.e. trial start, digging) using MATLAB (MathWorks). In order to classify neuronal representations of different task-related variables, we performed Receiver-Operating-Characteristic (ROC) analysis (*Green and Swets, 1966*) on the activity of each neuron prior to choice. Calcium traces were z-scored on a per-neuron basis across the entire session. For each neuron, switch representation was defined as significant calcium responses between early (trial-and-error rule learning) and late (rule-following) trials during a pre-choice time window ($-5$–$0$ s). Similar analysis was performed to classify trial history encoding, comparing calcium activity during the same time window ($-5$–$0$ s to from choice) after correct trials against after incorrect trials; and choice encoding, comparing calcium activity ($-5$–$0$ s from choice) when the upcoming choice on the current trial is correct or incorrect. A neuron was classified as responsive if its activity showed a significant difference ($p<0.05$) between two conditions within the defined time window in the ROC analysis.

## Dimensionality reduction

We concatenated neuronal activity across recordings, and constructed population vector in the early and late states by averaging calcium signals from all recorded neurons in all trials of a given state (in 50 ms bins, without overlap) over a period of ten seconds centered at the choice point (digging). These calcium values were extracted from 80% of recorded neurons randomly selected. This process was reiterated 20 times to account for the inherent variability in the dynamics of the population vector. The resultant high-dimensional trajectories were smoothed and embedded into a lower-dimensional space through principal component analysis (PCA). The explained variance was calculated for the first six principal components, which collectively accounted for over 80% of the total variance. Projections into a low-dimensional space (n=3) were generated for visualization purposes. Additionally, to evaluate the degree of similarity between state vectors, Pearson's correlation was computed for the time series of individual principal components. Subsequently, these correlation coefficients were averaged to derive an overall measure of vector similarity.

## Hidden Markov model

Following prior work (e.g. *Bagi et al., 2022; Recanatesi et al., 2022; Mazzucato et al., 2015*), we assume population neuronal activity can be clustered into two distinct (hidden) states, corresponding to the early learning state and the late learned state as observed in mouse behavior (*Figure 2e*). For

each trial, calcium activity from simultaneously recorded neurons was segmented into non-overlapping 50 ms windows and averaged over the 5 s period preceding the animal's choice (digging). Principal component analysis (PCA) was applied to identify low-dimensional representation of population activity of each trial. Based on these PCs, K-means was applied to group trials into clusters, initializing the parameters for HMM. The core assumption was that activity vectors corresponding to the same behavioral state would cluster together in the neuronal state space. An HMM was then fitted to estimate emission and transition probabilities between states. To ensure robustness, the clustering and modeling process was repeated 1000 times, with each iteration consisting of a randomly selected 40% of neurons. Model parameters were optimized using the Baum-Welch algorithm on 90% of the data, and performance was tested on the remaining 10% using the Viterbi algorithm (*Bishop, 2007*) to infer the most likely sequence of hidden states. To account for the potential confound that different number of trials affect model performance (LC inhibition typically required more trials than the control condition), we used a bootstrapping method to balance trial numbers. Specifically, we matched the total number of trials in each session to the highest possible number (31 trials).

## Generalized linear model

We conducted a logistic regression analysis on the population vectors to predict current trial outcomes (correct or incorrect) based on population activity patterns. To construct the population vector for each session, we initially computed the average activity of all recorded neurons in the 5 s window prior to choice. We then randomly selected 40% of neurons and applied principal component analysis. The first three principal components were retained as predictors for the regression model. We also matched trial numbers, following the methodology described earlier for HMM. To address variability and ensure robustness, we conducted 1000 bootstrap procedures. Subsequently, we partitioned 90% of the dataset for model training and tested the model on the remaining 10% of unseen data. A threshold of 0.5 was used to binarize the model probability. Values above 0.5 were assigned a label of 1 (correct choice), while values below 0.5 were assigned a label of 0 (incorrect choice). Model accuracy was assessed by comparing the actual behavioral sequence with model predicted sequence.

We note that incorrect choices likely reflect the early rule learning state, and correct choices likely reflect the late rule acquisition state. Thus, the two measurements of behavior, namely state change, and trial-by-trial choice, are not completely orthogonal to each other.

## Statistical analysis

Data were reported as mean ± SEM unless otherwise noted. We did not use statistical methods to predetermine sample sizes. Sample sizes were similar to those reported in the field. We assigned mice to experimental groups arbitrarily, without randomization or blinding. Unless otherwise noted, statistical tests were two-tailed Wilcoxon rank-sum when sample sizes were >7. When sample sizes were ≤7, two-tailed t-tests were used.

## Acknowledgements

We thank Shaorong Ma for helping with the behavioral paradigm, and Edward Zagha, Martin Riccomagno, and Sachiko Haga-Yamanaka for commenting on the manuscript. NEZ was supported by NIH grant R00DA047419. HY was supported by Klingenstein-Simons Fellowship Awards in Neuroscience and NIH grants R01NS107355 and R01NS112200.

## Additional information

### Competing interests

Natalie Zlebnik: Reviewing editor, *eLife*. The other authors declare that no competing interests exist.

## Funding

| Funder | Grant reference number | Author |
|---|---|---|
| National Institute of Neurological Disorders and Stroke | R01NS107355 | Hongdian Yang |
| National Institute of Neurological Disorders and Stroke | R01NS112200 | Hongdian Yang |
| National Institute on Drug Abuse | R00DA047419 | Natalie Zlebnik |

The funders had no role in study design, data collection and interpretation, or the decision to submit the work for publication.

## Author contributions

Marco Nigro, Formal analysis, Investigation, Methodology, Writing – original draft; Lucas Silva Tortorelli, Investigation; Machhindra Garad, Validation, Investigation; Natalie Zlebnik, Validation; Hongdian Yang, Conceptualization, Data curation, Formal analysis, Supervision, Funding acquisition, Writing – original draft, Writing – review and editing

## Author ORCIDs

Hongdian Yang ⓘ https://orcid.org/0000-0002-5203-9519

## Ethics

All experimental procedures were approved by the UC Riverside Animal Care and Use Committee (AUP20220030). All surgery was performed under isoflurane anesthesia, and every effort was made to minimize suffering.

Reviewer #3 (Public review): https://doi.org/10.7554/eLife.105911.4.sa1
Author response https://doi.org/10.7554/eLife.105911.4.sa2

# Additional files

## Supplementary files

Supplementary file 1. The fraction of specific groups of task-encoding neurons in individual mice from the control (n=4) and test (n=4) groups.

MDAR checklist

## Data availability

Source data contain the numerical data used to generate the figures.

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
