## [Editor Report · eLife Assessment]

This study presents a **valuable** finding on how the locus coeruleus modulates the involvement of medial prefrontal cortex in set shifting using calcium imaging in mice. The evidence supporting the claims was viewed as **solid** in revealing the dynamics and potential mechanisms supporting extradimensional shifts. The work is of broad interest to those studying flexible cognition.

---

## [Referee Report · Reviewer #3 (Public review)]

Summary:

Nigro et al examine how the locus coeruleus (LC) influences the medial prefrontal cortex (mPFC) during attentional shifts required for behavioral flexibility. Specifically, they propose that LC-mPFC inputs enable mice to shift attention effectively from texture to odor cues to optimize behavior. The LC and its noradrenergic projections to the mPFC have previously been implicated in this behavior. The authors further establish this by using chemogenetics to inhibit LC terminals in mPFC and show a selective deficit in extradimensional set shifting behavior. But the study's primary innovation is the simultaneous inhibition of LC while recording multineuron patterns of activity in mPFC. Analysis at the single neuron and population levels revealed broadened tuning properties, less distinct population dynamics, and disrupted predictive encoding when LC is inhibited. These findings add to our understanding of how neuromodulatory inputs shape attentional encoding in mPFC and are an important advance. There are some methodological limitations and/or caveats that should be considered when interpreting the findings and these are described below.

Strengths:

The naturalistic set-shifting task in freely-moving animals is a major strength, and the inclusion of localized suppression of LC-mPFC terminals builds confidence in the specificity of the behavioral effect. Combining chemogenetic inhibition of LC while simultaneously recording neural activity in mPFC with miniscopes is state-of-the-art. The authors apply analyses to population dynamics, in particular, that can advance our understanding of how the LC modifies patterns of mPFC neural activity. The authors show that neural encoding at both the single cell level and the population level are disrupted when LC is inhibited. They also show that activity is less able to predict key aspects of the behavior when the influence of LC is disrupted. This is quite interesting and adds to a growing understanding of how neuromodulatory systems sharpen tuning of mPFC activity.

Weaknesses:

Weaknesses are mostly minor, but there are some caveats that should be considered. First, the authors use a DBH-Cre mouse line and provide histological confirmation of overlap between HM4Di expression and TH immunostaining. While this strongly suggests modulation of noradrenergic circuit activity, the results should be interpreted conservatively as there is no independent confirmation that norepinephrine (NE) release is suppressed and these neurons are known to release other neurotransmitters and signaling peptides. In the absence of additional control experiments, it is important to recognize that effects on mPFC activity may or may not be directly due to LC-mPFC NE.

Another caveat is that the imaging analyses are entirely from the extradimensional shift session. Without analyzing activity data from the intradimensional shift (IDS) session, one cannot be certain that the observed changes are to some feature of activity that is specific to extradimensional shifts. Future experiments should examine animals with LC suppression during the IDS as well, which would show whether the observed effects are specific to an extradimensional shift and might explain behavioral effects.

Comments on revisions:

The authors overall do a nice job of addressing reviewer comments, and I believe the manuscript is significantly improved.

---

## [Author Response]

The following is the authors’ response to the previous reviews

We thank the reviewers and editors for the second round of peer review. Following the editorial assessment and specific review comments, we now present new results to compare EDS and IDS behavior, and use conventional standard for reporting statistics. We also request to simplify the manuscript title to be ‘Locus coeruleus modulation of prefrontal dynamics during attentional switching in mice’.

**Public Reviews:**

**Reviewer #1 (Public review):**
In their response to reviewers, the authors say "We report p values using 2 decimal points and standard language as suggested by this reviewer". However, no changes were made in the manuscript: for example, "P = 4.2e-3" rather than "p = 0.004".

We apologize for this misunderstanding. We initially interpreted this comment as reporting two non-zero digits in p values. We now have corrected this in the revision. We also follow the editorial recommendation and use a standard convention to report statistics (e.g., p = 0.03, t(7) = -2.8).

In their response to the reviewers, they wrote: "Upon closer examination of the behavioral data, we exclude several sessions where more trials were taken in IDS than in EDS." If those sessions in which EDSIDS. Most problematic is the fact that the manuscript now reads "Importantly, control mice (pooled from Fig. 1e, 1h, Supp. Fig. 1a, 1b) took more trials to complete EDS than IDS (Trials to criterion: IDS vs. EDS, 10 {plus minus} 1 trials vs. 16 {plus minus} 1 trials, P < 1e-3, Supp. Fig. 1c), further supporting the validity of attentional switching (as in Fig. 1c)" without mentioning that data has been excluded.

Editor raised a similar concern. We apologize for this oversight, which was due to miscommunication within the lab. We have now revised the manuscript to include all data points without any exclusion in Fig. 1e, 1h, and Supp. Fig. 1a-c. By pooling all data without any exclusion, control mice readily took more trials to complete EDS than IDS, supporting the validity of attentional switching (Trials to criterion: IDS vs. EDS, 11 ± 1 trials vs. 15 ± 1 trials, p = 0.006, Supp. Fig. 1c).

The exclusion we initially meant to perform was to exclude sessions where task performance in IDS was beyond 95% threshold inferred from the naïve control group (15 trials, Fig. 1c). Exclusions are now explicitly described. Of note, including or excluding these sessions does not change any of the conclusions presented in our manuscript. We have added this analysis in Supp. Fig. 1d and the results remain robust (Supp. Fig. 1d). This panel could be removed if deemed unnecessary by the reviewers.

**Reviewer #3 (Public review):**
The authors overall do a nice job of addressing reviewer comments, and I believe the manuscript is significantly improved. Congratulations!

We thank you for this positive assessment.

Weaknesses are mostly minor, but there are some caveats that should be considered. First, the authors use a DBH-Cre mouse line and provide histological confirmation of overlap between HM4Di expression and TH immunostaining. While this strongly suggests modulation of noradrenergic circuit activity, the results should be interpreted conservatively as there is no independent confirmation that norepinephrine (NE) release is suppressed and these neurons are known to release other neurotransmitters and signaling peptides. In the absence of additional control experiments, it is important to recognize that effects on mPFC activity may or may not be directly due to LC-mPFC NE.

We agree with this comment, and now further discuss this limitation in Discussion, line 255-259:

“However, it is important to note that LC-NE neurons can co-release other neurotransmitters, such as dopamine and neuropeptides[73,75,76]. In the absence of further control experiments to confirm the suppression of NE release, the observed effects on mPFC may or may not be directly due to NE. Future studies are needed to better delineate the involvement of specific neurotransmitters, cell types and receptors in flexible decision making.”

Another caveat is that the imaging analyses are entirely from the extradimensional shift session. Without analyzing activity data from the intradimensional shift (IDS) session, one cannot be certain that the observed changes are to some feature of activity that is specific to extradimensional shifts. Future experiments should examine animals with LC suppression during the IDS as well, which would show whether the observed effects are specific to an extradimensional shift and might explain behavioral effects.

We also agree with this comment, and have thought about this. Technically, IDS has low trial numbers, especially incorrect trials, limiting the power of statistical comparisons. Conceptually, since in our paradigm EDS is always the last stage, comparing neural signals in EDS with previous stages may be confounded by the order of learning. That is, whether the observed differences in mPFC activity were due to mPFC responding to different rules, or due to mPFC responses over time/learning. We now discuss this point in Discussion, line 291-295:

“Another limitation in the current study is that neurophysiological analyses were entirely from EDS. Without comparing with other task stages (e.g., REV, IDS), it is uncertain to what extent the observed neuronal changes are specific to EDS. Future experiments should examine the behavioral and neurophysiological effects with LC inhibition to determine the specificity of LC-NE modulation of the mPFC during attentional switching.”

We are also actively collecting additional data to address this point, which requires considerable efforts. We hope to report our findings in a follow up study.